# Empowering Psychotherapy with Large Language Models: Cognitive Distortion Detection through Diagnosis of Thought Prompting

**Zhiyu Chen[1], Yujie Lu[2], and William Yang Wang[2]**
[1]Carnegie Mellon University
[2]University of California, Santa Barbara
`zhiyuc@andrew.cmu.edu`,
`{yujielu, william}@cs.ucsb.edu`

## Abstract

Mental illness remains one of the most critical public health issues of our time, due to the severe scarcity and accessibility limit of professionals. Psychotherapy requires high-level expertise to conduct deep, complex reasoning and analysis on the cognition modeling of the patients. In the era of Large Language Models, we believe it is the right time to develop AI assistance for computational psychotherapy. We study the task of cognitive distortion detection and propose the **Diagnosis of Thought (DoT)** prompting. DoT performs diagnosis on the patient's speech via three stages: *subjectivity assessment* to separate the facts and the thoughts; *contrastive reasoning* to elicit the reasoning processes supporting and contradicting the thoughts; and *schema analysis* to summarize the cognition schemas. The generated diagnosis rationales through the three stages are essential for assisting the professionals. Experiments demonstrate that DoT obtains significant improvements over ChatGPT for cognitive distortion detection, while generating high-quality rationales approved by human experts.

## 1 Introduction

About one in eight people worldwide suffer from mental disorders (World Health Organization, 2022). However, such mental health conditions are severely underserved, due to a number of reasons including the scarcity of mental health professionals, poor quality of services, unaffordable cost, and social stigma (White and Dorman, 2001; Sharma et al., 2020b). Treatment coverage for mental health service use ranges from 33% in high-income locations to only 8% in low- and lower middle-income countries (Moitra et al., 2022). A recent study from the American Psychological Association (APA)[1] found that six in ten psychologists "no longer have openings for new patients."

To mitigate such situations, there have been consistent efforts on developing automated systems for mental health support, such as sentiment analysis (Rathje et al., 2023) and empathetic chatbots (Welivita et al., 2021; Sharma et al., 2020b; Saha et al., 2022). However, existing works mostly take shallow attempts in a heuristic manner, e.g., analyzing emotions and generating comforting responses. There is still a significant gap for such systems to contribute to real professional psychotherapy, which requires deep studies of the patient's thinking patterns, the establishment of cognition models, and the methods to reconstruct the cognition models. These procedures constitute the core pillars in common classic therapy paradigms like cognitive-behavior therapy (CBT) (Rothbaum et al., 2000; Wright et al., 2017; Beck, 2020) and acceptance and commitment therapy (ACT) (Harris, 2006; Hayes et al., 2011). Most data sources recording the interactions between the patients and licensed professionals are confidential, making it even more challenging to build professional assistance for psychotherapy.

Recent progress in Large Language Model (LLM) development uncovers its astonishing ability in various textual reasoning tasks in zero-shot setting (Kojima et al., 2022; Bang et al., 2023; Chen, 2022). For the psychology domain, ChatGPT and GPT-4 present very promising performance in the classic Sally-Anne test (Baron-Cohen et al., 1985; Bubeck et al., 2023), evaluating the basic theory of mind capabilities to attribute mental states such as beliefs, emotions, desires, etc. We believe it is promising to further exploit such ability to complex mental reasoning and analysis. It is the right time to start developing professional, targeted, and systematic AI assistance for psychotherapy.

In this paper, we take the first step by studying the task of **cognitive distortion detection**, the first core procedure in cognitive behavior therapy (CBT) (Beck, 2020). Inspired by how psychother-

---

[1]https://www.theguardian.com/society/2022/nov/21/therapist-shortage-us-psychologists-pandemic

| Cognitive Distortion Type | Interpretation | Example Distorted Speech |
|---|---|---|
| Personalization | Personalizing or taking up the blame for a situation, that in reality involved many factors and was out of the person's control. | My son is pretty quiet today. I wonder what I did to upset him. |
| Mind Reading | Suspecting what others are thinking or what are the motivations behind their actions. | My house was dirty when my friends came over, they must think I'm a slob! |
| All-or-nothing thinking | Looking at a situation as either black or white or thinking that there are only two possible outcomes to a situation. | If I cannot get my Ph.D., then I am a total failure. |

Table 1: Three example common cognitive distortion types, taken from (Beck, 2020; Shreevastava and Foltz, 2021).

apy professionals perform nuanced diagnosis over the patient's speech, we propose the **Diagnosis of Thought (DoT) prompting**. In DoT, we diagnose the patient's speech through three stages: (1) subjectivity assessment, (2) contrastive reasoning, and (3) schema analysis. In subjectivity assessment, we distinguish the patient's subjective thoughts from the objective facts; In contrastive reasoning, we elicit the reasoning processes supporting and opposing the patient's thoughts; Finally, in schema analysis, we summarize the underlying thought schema and map it to the cognitive distortion types. We conduct comprehensive experiments using the recent top-performing LLMs. In zero-shot setting, DoT obtains over 10% and 15% relative improvements for distortion assessment and classification, respectively, on ChatGPT. Meanwhile, the generated rationales through the three stages grant full interpretability for the diagnosis process, whose quality is further approved by human experts.

We unveil the great potential of empowering professional psychotherapy with LLM. This exploration serves as a catalyst for a larger initiative; we extend an invitation to both AI and psychotherapy communities to come together in a collaborative effort. Our ultimate goal is to construct professional, safe, AI-driven assistance that can substantially enhance mental health support systems.

## 2 Cognitive Distortion Detection

Cognitive Behavior Therapy (CBT) is a well-established therapy paradigm primarily on depression and anxiety disorders (Beck, 2020). In CBT, given the patients' speech or written content, we establish the cognitive model by building the interactions among the situations, thoughts, and emotions. Patients with mental disorders, such as depression or anxiety, tend to form negative thoughts very rapidly and unconsciously, leading to negative emotions which further strengthen their overall negative views and beliefs about the world.

To break this vicious circle, in a typical CBT process, the first core step is to identify those mal-adaptive negative thoughts and summarize their underlying schemas, formally known as cognitive distortions. There are generally 10-20 common, well-studied types of cognitive distortions. We present a few types with examples in Table 1, and refer the readers to Appendix A for the full list. Once accurately identify these cognitive distortions, CBT therapists will guide the patients to justify and correct these distortions, so as to gradually reconstruct their cognition models.

In the real psychotherapy process, there's a significant amount of textual information including therapeutic conversations and diaries, etc. Such information is often long, highly fragmented, and disorganized, containing multiple types of distortions beyond the toy examples in Table 1. The task of cognitive distortion detection aims to automatically detect the distortion types given such textual information from the patients, in order to assist therapists to enhance their efficiency and productivity. Meanwhile, such detectors can also potentially serve as self-assisting tools for the patients to diagnose their thoughts and conduct CBT practice, upon meeting the robustness and safety requirements. Formally, cognitive distortion detection consists of two steps: 1) **Distortion assessment** to predict whether the given speech contains cognitive distortions, as a binary classification problem; and 2) **Distortion classification** to predict the specific distortion types, as a multiclass classification problem.

## 3 The Diagnosis of Thought Prompting

On discussing with psychotherapy professionals, we identify the following key stages to diagnose the patient's speech to detect cognitive distortions. **Subjectivity Assessment.** The patient's speech consists of a mixture of reality (objective facts) and interpretations/opinions (subjective thoughts). In order to perform deep analysis of distorted thinking, we first need to find out which parts of the speech are objective facts and which parts are subjective thoughts. After such an assessment, we summa-

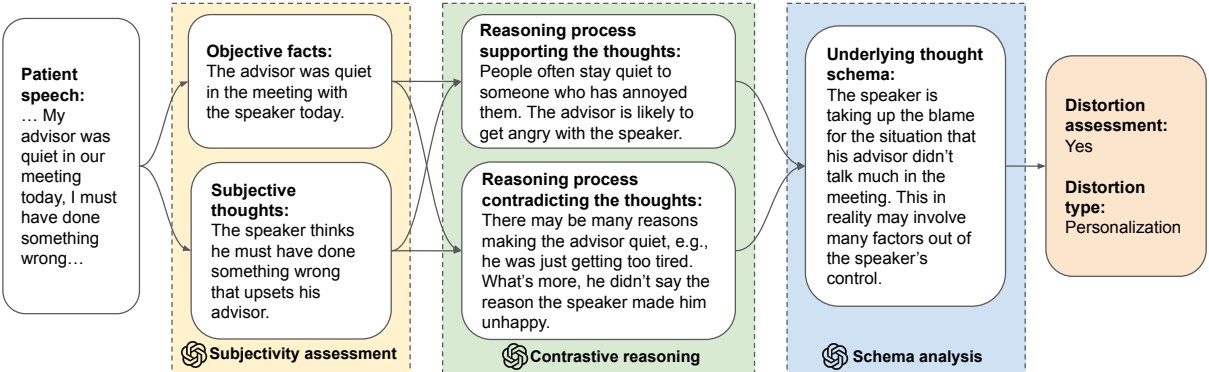

Figure 1: The Diagnosis of Thought framework. We strategically prompt the LLM to go through the three diagnosis stages: subjectivity assessment, contrastive reasoning, and schema analysis.

rize the objective facts into the situations as the evidence base to diagnose the subjective thoughts.

**Contrastive Reasoning.** This stage aims to discover how the patient ascertains the veracity of their subjective thoughts. Based on the situation, we deduct the reasoning processes that supports and contradicts the patient's thoughts respectively. By contrasting two different interpretations based on the same situation, we can identify the thought schemas more clearly.

**Schema Analysis.** This stage aims to study why the patient forms the specific reasoning process. The term "schema" refers to the cognitive structures that organize our knowledge, beliefs, and expectations. Understanding what schemas a patient is relying on can reveal much about their cognitive mode and distortions.

We propose the Diagnosis of Thought (DoT) prompting, guiding the LLM through the above three stages to diagnose the patient's speech. Figure 1 illustrates our framework. We sequentially prompt LLM with three questions for the three stages. After the LLM finishes the generation for all stages, we prompt it with another two questions asking distortion assessment and classification. See Appendix B for all the prompts we use.

With DoT prompting, we obtain fully interpretable diagnosis rationales for detecting cognitive distortions. Such interpretability is vital for professionals in real applications. Serving as the diagnosing assistance tool, the diagnosis rationales are essential for the professionals to justify the results. More importantly, they can potentially learn the nuanced thought patterns and schemas derived from the patient's speech through the rationales. This is crucial for the professionals to establish the patient's cognition models more efficiently from the vast amount of speech.

## 4 Experimental Results

### 4.1 Dataset and experimental settings

We experiment on the cognitive distortion detection dataset proposed by Shreevastava and Foltz (2021), which is annotated by experts based on the Therapist QA dataset[2]. The dataset consists of 2,531 examples of patient speech annotated with ten common types of cognitive distortions, as specified in Appendix A. 63.1% of the examples have cognitive distortions, which are annotated with the two dominant ones. We follow the same train-test split in (Shreevastava and Foltz, 2021). For automatic evaluations, we report F-1 for distortion assessment and weighted F-1 for distortion classification.

### 4.2 Automatic Evaluation Results

We experiment using three of the recent representative LLMs: ChatGPT (gpt-3.5-turbo)[3], Vicuna[4], and GPT-4 (OpenAI, 2023). For all methods, we first prompt with a general instruction specifying the task and the target distortion types. We compare our DoT prompting with 1) Directly generating the results, and 2) Zero-Shot CoT prompting (ZCoT) (Kojima et al., 2022). For Vicuna, both ZCoT and DoT exceed the token limits for many examples; we omit the results. We average over five runs and report mean and standard deviation for all experiments. Table 2 shows our main experiment results. As we can see, under the zero-shot setting, due to the challenge of this task, Vicuna and Chat-GPT still fall behind the supervised full-training models. The proposed DoT prompting significantly

[2]https://www.kaggle.com/arnmaud/therapist-qa
[3]https://openai.com/blog/chatgpt
[4]https://lmsys.org/blog/2023-03-30-vicuna/

| Methods | Distortion Assessment (F-1) | Distortion Classification (Weighted F-1) |
|---|---|---|
| Full training[*] | 75.0 | 24.0 |
| Vicuna | $73.81_{0.95}$ | $11.23_{0.78}$ |
| ChatGPT | $73.47_{0.58}$ | $19.24_{1.00}$ |
| ChatGPT + ZCoT | $77.10_{1.21}$ | $20.21_{1.02}$ |
| ChatGPT + DoT | $81.19_{0.11}$ | $22.25_{0.70}$ |
| GPT-4 | $83.04_{0.51}$ | $33.86_{0.83}$ |
| GPT-4 + ZCoT | $81.97_{1.21}$ | $33.22_{1.36}$ |
| GPT-4 + DoT | $82.77_{0.81}$ | $34.64_{1.40}$ |

Table 2: Main Results (Standard deviation in subscript). ([*]) results copied from (Shreevastava and Foltz, 2021).

| Methods | Distortion Assessment (F-1) | Distortion Classification (Weighted F-1) |
|---|---|---|
| ChatGPT | $73.47_{0.58}$ | $19.24_{1.00}$ |
| ChatGPT + S1 | $79.62_{1.12}$ | $18.72_{1.95}$ |
| ChatGPT + S1 + S2 | $80.70_{0.48}$ | $20.11_{1.02}$ |
| ChatGPT + S1 + S2 + S3 | $81.19_{0.11}$ | $22.25_{0.70}$ |

Table 3: Ablation studies on each stage of DoT: we denote Subjectivity Assessment as S1, Contrastive Reasoning as S2, and Schema Analysis as S3.

boosts the performance of ChatGPT, with the distortion assessment score surpassing the full-training performance and the distortion classification score approaching the full-training performance. The improvement over ZCoT demonstrates the superiority of our proposed strategic stages compared with rationale generation with no guidance. Astonishingly, GPT-4 surpasses the full-training performance even by a large margin for distortion classification. On the one hand, this may demonstrate the powerful ability of GPT-4 on this task - it can already reach a potential upper bound regardless with DoT promoting or not; On the other hand, since the dataset we use is publicly released on Kaggle and its original data source is from online forums, we do not exclude the possibility of data contamination.

**Ablation Studies and Analysis.** To better understand how each stage of DoT contributes to overall performance, we conduct ablation studies using ChatGPT shown in Table 3. All three stages obtain improvements for distortion assessment and distortion classification. For distortion assessment, subjectivity assessment (S1) gains the largest improvement, as separating objective facts and subjective thoughts is a strong trigger to assess distortions. For distortion classification, Schema Analysis (S3)

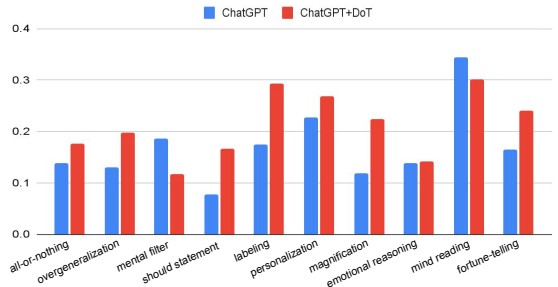

Figure 2: Distortion classification per class F-1.

| Methods | Quality | Subjectivity Assessment | Contrastive Reasoning | Schema Analysis |
|---|---|---|---|---|
| ChatGPT +DoT | Comprehensive | 68.5% | 45.0% | 50.5% |
|  | Partially good | 24.5% | 30.5% | 27.0% |
|  | Invalid | 7.0% | 24.5% | 22.5% |
| GPT4 +DoT | Comprehensive | 84.5% | 69.5% | 76.5% |
|  | Partially good | 13.5% | 25.5% | 18.5% |
|  | Invalid | 2.0% | 5.0% | 5.0% |

Table 4: Human evaluation results for diagnosis rationales.

obtains large improvement, as the summarized underlying thought schema can match the definition of the distortion types more accurately. We also analyze the distortion classification results for each distortion type, with results shown in Figure 2.

### 4.3 Human Evaluation Results

As there is no reference available for the diagnosis rationales, we employ psychotherapy experts to assess the quality of the generated rationales. We hired psychotherapy professionals from Up-Work[5], e.g., certified clinicians, counseling psychology Ph.D. students, etc. We discussed with each hire to reach an agreement on the payment following a fair wage rate.

Specifically, we present the patient's speech, all the prompts, and the generated rationales for all stages to human experts. For the generated rationales of each stage, we instruct the experts to choose between: 1) Comprehensive. (Correct and comprehensive.); 2) Partially good. (Reasonable but not comprehensive) 3) Invalid. (Not reasonable.) Table 4 shows the evaluation results on 100 examples for DoT over ChatGPT and GPT-4. We employ two experts for each evaluation and report the average; The agreement rates (the percentage of examples the two experts gave the same rating) for both evaluations are over 80%. The diagnosis rationales generated for all stages shows decent

---

[5]www.upwork.com

quality verified by the experts. See Appendix C for more generation examples.

## 5   Related Work

Due to the verbal and textual nature of psychotherapy procedures, there have been consistent efforts to employ NLP techniques to assist the mental health domain (Althoff et al., 2016; Abd-Alrazaq et al., 2021, 2019; Valizadeh and Parde, 2022). However, with the knowledge gap between the two communities, most existing works take shallow attempts without deep investigation of the professional psychotherapy knowledge. A majority of the previous studies targets identification of common mental health issues, such as depression and anxiety, from textual contents (Cohan et al., 2018; MacAvaney et al., 2018; Harrigian et al., 2020; Ji et al., 2022; Zanwar et al., 2023; Juhng et al., 2023). Another major category of works study therapeutic conversations with a focus on emotional/empathetic supports (Halder et al., 2017; Sharma et al., 2020a; Atapattu et al., 2022; Mishra et al., 2023), and discourse structures (Cao et al., 2019; Hsu et al., 2023). A few more recent works have started to investigate deeper professional psychotherapy knowledge, e.g., cognitive distortion detection in CBT (Shreevastava and Foltz, 2021; Ding et al., 2022; Lybarger et al., 2022), identifying and reframing unhelpful thoughts (Ziems et al., 2022; Maddela et al., 2023). Early famous system Eliza[6] took the initial attempts to emulate a Rogerian psychotherapist. Due to its rule-based nature, the responses are often reflections or rephrasings of the user's statements. If users deviate too much from expected inputs or probe its capabilities, the program may produce irrelevant or nonsensical responses.

Recent progress in large language model reveals that ChatGPT and GPT-4 present very promising performance in the classic Sally-Anne test (Baron-Cohen et al., 1985; Bubeck et al., 2023), evaluating the basic theory of mind capabilities to attribute mental states such as beliefs, emotions, desires, etc. However, latter works also point out the robustness issue of such ability (Shapira et al., 2023). In our work, we are inspired by such enhanced ability and investigate the application of diagnosing patients' thoughts in psychotherapy. Our experiments on the real dataset, not anecdotal examples, show very promising performance both in auto-

---

[6] https://en.wikipedia.org/wiki/ELIZA

matic and human expert evaluation. We believe it is a very important future research direction to investigate the general cognitive abilities of LLM and make connections to the field of psychology and cognitive science. Theory-wise, we are eager to explore to what extent the LLM can simulate the human cognitive functions, so as to determine the role that language plays in the overall human cognition. Application-wise, a straightforward and encouraging direction should be the integration for assisting mental health treatment, as the motivation and goal of this work.

## 6   Conclusion and Future Work

This work delved into the integration of large language models (LLMs) within the realm of psychotherapy. We introduced the Diagnosis of Thought (DoT) framework, which strategically prompts the LLM to produce diagnosis rationales pertinent to the detection of cognitive distortions. We believe that there exists substantial potential for leveraging LLMs within numerous facets of mental health support that are currently under-explored. Our findings, thus, not only illuminate a path towards more efficient therapy methods but also open doors for future investigations to push the boundaries of AI's role in mental health treatment.

In the domain of AI-enhanced psychotherapy, particularly when harnessing the power of LLMs, there are paramount challenges surrounding safety, robustness, and ethical responsibilities. LLM is known to generate hallucinations and biases. While one potential usage of our system is to serve as a self-diagnosing/assisting tool for patients, e.g., to help the patients to recognize their own thinking patterns and learn how the cognitive distortions developed, such tools should not be deployed for direct patient use without the supervision of professionals. As we navigate the potential of LLM in psychotherapy, our foremost priority should remain in building systems that are safe and responsible. Concurrently, the formulation of robust ethical guidelines tailored to this new era becomes indispensable.

## Limitations

One of the biggest obstacles to studying AI for psychotherapy is the lack of available, high-quality data. Most of the datasets recording the information of the interactions between the patients and licensed professionals are confidential due to pri-

vacy concerns. The dataset used in this work is the only one we can find publicly available for cognitive distortion detection, as its data source is from public online forums. In addition to the patient's speech, the patient's demographic information also plays an important role in analyzing the thought process leading to cognitive distortions. In this dataset, some of the patient speech include a few demographic details, but pretty minimal. Beyond the task of cognitive distortion detection, the same data constraint issue exists for all other therapy stages and paradigms. This also motivates us to work on building privacy-preserved systems as a crucial future direction.

## Ethics Statement

The system built in this work are never meant to replace psychotherapy professionals: all system-produced results need to be verified by licensed professionals. The system should not be deployed for direct patient use without the supervision of licensed professionals.

There has been a lot of discussions regarding the use of LLMs for AI-driven therapy purpose. Chatbots will find their most effective role in medicine within the realm of mental health, an opinion[7] quoted from Thomas Insel, former director of the National Institute of Mental Health. In the meantime, we should always take caution for such development. Our foremost priority should be building safe and responsible applications while establishing ethical and regulation standards for this new era. Under the current status, the partnership between professionals and AI systems is a promising direction for the goal to ease the burden of professionals, which is the motivation of this work.

"Diagnosis of Thought" is an innovative approach that harnesses the capabilities of large language models to enhance the field of psychotherapy. It is imperative to understand that this tool is not designed for censorship or any form of invasive surveillance. Instead, it aims to augment the skills of therapists by providing them with advanced, data-driven insights into their clients' thoughts and emotions. Through the efficient processing of linguistic cues and patterns, these models aid mental health professionals in formulating more precise diagnoses and crafting tailored therapeutic interventions that could significantly enhance the well-

being of individuals seeking help.

For all the annotations and human evaluations presented in this work, we hired psychotherapy professionals from UpWork[8], e.g., certified clinicians, counseling psychology Ph.D. students, etc. We discussed with each hire to reach an agreement on the payment following a fair wage rate. The average hourly rate for all the experts is $80.

This project is approved by our Institutional Review Board (IRB). The data annotation is classified as exempt by our IRB.

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

## Appendix A: Cognitive Distortion Types

We list all the ten cognitive distortion types studies in this work following·(Shreevastava and Foltz, 2021) in Table 5. Note that these ten types being studied are some common, well-studied ones. As psychotherapy develops, more fine-grained and new distortion types are expected to be unveiled.

For the cognitive distortion detection dataset proposed by Shreevastava and Foltz (2021), there are 2,531 examples in total. We follow the original 80% - 20% train-test split. The input patient speech has an average length of 167.3 tokens, and the ten distortion types are roughly equally distributed.

## Appendix B: Prompt Details

For all our experiments without using DoT, we first prompt the LLM with the following general instructions:

Given a speech of a patient, our task is to 1) identify if there is cognitive distortion in the speech; 2) Recognizing the specific types of the cognitive distortion. Here we consider the following common distortions: (followed by the descriptions and examples for all ten prompts in the dataset metadata of (Shreevastava and Foltz, 2021).

For all our experiments using DoT, we first prompt the LLM with the following general instructions:

Given a speech of a patient, our task is to 1) finish a few diagnose of thought questions to analyze the thought patterns of the patient. Then based on the diagnose of thought analysis, 2) identify if there is cognitive distortion in the speech; 3) Recognizing the specific types of the cognitive distortion. Here we consider the following common distortions: (followed by the descriptions and examples for all ten prompts in the dataset metadata of (Shreevastava and Foltz, 2021).

Then for methods without using DoT, we prompt the following two questions:

Please first answer: if there is cognitive distortion in the speech; Answer 'yes' or 'no'; Please then answer: Recognizing the specific types of the cognitive distortion in the speech. There may be

| Cognitive Distortion Type | Interpretation | Example Distorted Speech |
|---|---|---|
| Personalization | Personalizing or taking up the blame for a situation, that in reality involved many factors and was out of the person's control. | My son is pretty quiet today. I wonder what I did to upset him. |
| Mind Reading | Suspecting what others are thinking or what are the motivations behind their actions. | My house was dirty when my friends came over, they must think I'm a slob! |
| Overgeneralization | Major conclusions are drawn based on limited information. | Last time I was in the pool I almost drowned, I am a terrible swimmer and should not go into the water again. |
| All-or-nothing thinking | Looking at a situation as either black or white or thinking that there are only two possible outcomes to a situation. | If I cannot get my Ph.D., then I am a total failure. |
| Emotional reasoning | Letting one's feeling about something overrule facts to the contrary. | Even though Steve is here at work late every day, I know I work harder than anyone else at my job. |
| Labeling | Giving someone or something a label without finding out more about it/them. | My daughter would never do anything I disapproved of. |
| Magnification | Emphasizing the negative or playing down the positive of a situation. | My professor said he made some corrections on my paper, so I know I'll probably fail the class. |
| Mental filter | Placing all one's attention o, or seeing only, the negatives of a situation. | My husband says he wishes I was better at housekeeping, so I must be a lousy wife. |
| Should statements | Should statements appear as a list of ironclad rules about how a person should behave, this could be about the speaker themselves or other. It is NOT necessary that the word 'should' or it's synonyms (ought to, must etc.) be present in the statements containing this distortion. | I should get all A's to be a good student. |
| Fortune-telling | As the name suggests, this distortion is about expecting things to happen a certain way, or assuming that thing will go badly. Counterintuitively, this distortion does not always have future tense. | I was afraid of job interviews so I decided to start my own thing. |

Table 5: Common cognitive distortion types and example speech, taken from (Beck, 2020; Shreevastava and Foltz, 2021).

one type of cognitive distortion or multiple types involved. If there are multiple types, please give the top 2 dominant ones. Please only give the distortion type names separated by comma.

For methods using DoT, before asking this two questions, we provide the following DoT prompts:

Based on the patient's speech, finish the following diagnosis of thought questions: 1. what is the situation? Find out the facts that are objective; what is the patient thinking or imagining? Find out the thoughts or opinions that are subjective. 2. what makes the patient think the thought is true or is not true? Find out the reasoning processes that support and do not support these thoughts. 3. why does the patient come up with such reasoning process supporting the thought? What's the underlying cognition mode of it?

## Appendix C: Generation Examples

We present some generation examples of GPT-3.5 and GPT-4 in Figure 3.

**Patient Message:** Lately I have been extremely unhappy for no apparent reason. I feel empty almost all the time. It is like nothing can give me genuine joy anymore. I also feel alone a lot, even though I do have friends and close friends. Whenever I try to imagine my future all I see is one full of obstacles. I have become so consumed in sadness and I do not know how to deal anymore. I feel so confused and I do not understand myself anymore.

**Diagnosis of Thought (ChatGPT):**

| | |
|---|---|
| **Objective Fact:** | The patient has been feeling extremely unhappy and empty lately. |
| **Subjective Thought:** | Nothing can give the patient genuine joy and that the future is full of obstacles. the patient feels confused and does not understand themselves anymore. |
| **Reasoning Process Supporting the Thoughts:** | The patient may believe that their thoughts are true because they are experiencing these feelings and emotions firsthand. |
| **Reasoning Process Contradicting the Thoughts:** | There may not be factual evidence to support the belief that nothing can give the patient genuine joy or that the future is full of obstacles. the patient may be using emotional reasoning to support their beliefs. |
| **Cognition Schema:** | The patient may be experiencing a cognitive mode of negativity bias, where they tend to focus on negative experiences and thoughts rather than positive ones. this may be influencing their reasoning processes, leading them to magnify the negative and filter out the positive aspects of their life and future. additionally, the patient may be struggling with depression, which can impact the way they perceive themselves and the world around them. |

**Diagnosis of Thought (GPT4):**

| | |
|---|---|
| **Objective Fact:** | The patient has been extremely unhappy recently. The patient feels empty and alone most of the time. he patient has friends and close friends. |
| **Subjective Thought:** | They believe nothing can bring them genuine joy anymore. They imagine their future full of obstacles. They feel consumed by sadness and confusion and no longer understand themselves. |
| **Reasoning Process Supporting the Thoughts:** | The persistent feeling of emptiness and lack of joy might make the patient believe that their situation is unchanging and will remain the same. They may be experiencing negative events in their lives, making it harder to feel hopeful about their future. Feeling alone even when surrounded by friends could cause them to think that their friendships are not fulfilling or meaningful. |
| **Reasoning Process Contradicting the Thoughts:** | Emotions and feelings are temporary and can change over time, so it is not guaranteed that the patient will always feel this way. The patient is aware that they do have friends, which suggests that some level of meaningful connection exists. Obstacles are a natural part of life, but they are not the only things present in one's future. positive events and achievements are also likely to occur. |
| **Cognition Schema:** | Emotional reasoning: they believe that their feelings of unhappiness, emptiness, and loneliness are true reflections of their life's reality. Mental filter: they are focusing on the negative aspects of their life and filtering out positive experiences or memories. Fortune-telling: they are assuming that their future will be full of obstacles based on their current emotions and thoughts. |

**Patient Message:** So I have no close friends at the moment probably I know some people enough just to say hi but no close one and whenever I make any they go away I don't know if its my personality the reason or the way I behave and the only friends I can gain are school friends since there is no other place I go to I can't really stand myself alone. Thanks for helping.

**Diagnosis of Thought (ChatGPT):**

| | |
|---|---|
| **Objective Fact:** | The patient has no close friends at the moment and feels lonely.The patient has made acquaintances but no close friends. |
| **Subjective Thought:** | The patient feels that they are the reason for their friends leaving and wonders if it's their personality or behavior causing the issue. they cannot stand being alone. |
| **Reasoning Process Supporting the Thoughts:** | The patient believes their thoughts to be true because they have not been able to maintain any close friendships and the pattern keeps repeating itself. they may be using past experiences to support this belief. |
| **Reasoning Process Contradicting the Thoughts:** | There may be other reasons for the loss of these friendships that the patient is not considering. |
| **Cognition Schema:** | The patient may be experiencing cognitive distortions such as overgeneralization by assuming that all friendships will end up the same way and mental filtering by focusing only on the negative aspects of their past friendships. the underlying cognition mode could be a negative self-image or low self-esteem. |

**Diagnosis of Thought (GPT4):**

| | |
|---|---|
| **Objective Fact:** | The patient currently does not have close friends and mainly interacts with school friends. they feel lonely and seek help. |
| **Subjective Thought:** | The patient believes that their personality or behavior might be the reason their friendships don't last, and they are limited in opportunities to make new friends. |
| **Reasoning Process Supporting the Thoughts:** | The patient might have experienced failed friendships and noticed a pattern of people leaving their life. this leads them to question if their personality or behavior is the cause. |
| **Reasoning Process Contradicting the Thoughts:** | The patient has not provided concrete examples or evidence that their personality or behavior is the reason behind the lack of close friendships. it is possible there are external factors or situational circumstances that led to the loss of friendships. |
| **Cognition Schema:** | The patient might be engaging in personalization and overgeneralization cognitive distortions. personalization because they seem to be taking the blame for friendships not lasting, assuming it's their fault due to their personality or behavior. overgeneralization because they are making a broad conclusion based on limited information (i.e., the assumption that their personality or behavior is the sole reason for the lack of close friends). this underlying cognition mode could stem from low self-esteem, fear of rejection, or previous negative experiences in social situations. |

Figure 3: Showcases of Diagnosis of Thought using ChatGPT and GPT4 given the Patient Message.