# OpenReview forum: "Empowering Psychotherapy with Large Language Models: Cognitive Distortion Detection through Diagnosis of Thought Prompting"
_EMNLP/2023/Conference — EMNLP 2023 Findings_

### Official Review · Reviewer_7GRj · 2023-08-02

**Soundness:** 2

**Excitement:**

2: Mediocre: This paper makes marginal contributions (vs non-contemporaneous work), so I would rather not see it in the conference.

**Paper Topic And Main Contributions:**

The proposed DoT approach addresses the pressing need for more accessible and efficient mental health support. By leveraging Large Language Models, it aims to provide valuable AI assistance to mental health professionals, potentially bridging the gap between supply and demand for psychotherapy services. By leveraging the power of LLMs, the approach demonstrates potential improvements over existing methods and provides valuable rationales for both professionals and patients.


**Reasons To Accept:**

1. The utilization of LLM in this context of mental healthcare could lead to faster and more accurate identification of cognitive distortions, enhancing the effectiveness of therapy sessions.
2. The DoT method involves a three-stage process for cognitive distortion detection, which proposed a good way of prompt engineering to elicit the capabilities of LLMs.

**Reasons To Reject:**

1. This paper is a case study that utilize LLMs to generate texts in the field of mental health. The main contribution is proposing a prompt based on CBT, which is good but not sufficiently novel to appear in such a competitive conference.
2. "Diagnosis" is kind of over-claim. There is no clear evidence that LLMs can really diagnose.

**Reproducibility:**

4: Could mostly reproduce the results, but there may be some variation because of sample variance or minor variations in their interpretation of the protocol or method.

**Reviewer Confidence:**

3: Pretty sure, but there's a chance I missed something. Although I have a good feel for this area in general, I did not carefully check the paper's details, e.g., the math, experimental design, or novelty.

---

> ### Author Rebuttal · Authors · 2023-08-28
>
> We thank the reviewer for the comments and suggestions. We would like to address your concerns as follows:
>
> ### 1. Regarding “...not sufficiently novel contribution”.
> Our proposed DoT framework comes from the consultation and study jointly with psychotherapy experts. The framework is concise and elegant - all stages in DoT address the key insights and challenges in diagnosing the patients’ thoughts without any redundancies. Each stage brings significant improvement for distortion assessment and classification, and the generated rationales are highly valuable for psychotherapy professionals to interpret, analyze, as well as learn more insights into the patients’ thought patterns. Furthermore, DoT, as well as its related work like "Chain of Thought (CoT)" (Wei et al., 2022), are essentially studies towards cognitive science and psychology for LLMs rather than prompt engineering. We explore the abilities that LLMs elicit like the way humans think and reason, and study how such cognition abilities emerged (Physics of LLM), or extrapolate such abilities into real-world applications (like this work). Our work is different from and goes beyond CoT in that we design strategies to elicit more complex and fine-grained cognition abilities based on psychotherapy domain knowledge. The advanced cognition abilities we demonstrate can potentially revolutionize the field of AI for psychotherapy.
>
> Given the critical condition of mental health issues and the severe shortage of therapists today, we believe it is imperative, or at least highly valuable to call on efforts for NLP in psychotherapy, which can benefit billions of lives. We believe the contribution of this work is far beyond a "case study" - with the proposed DoT method, we unveil the power of LLM in the deep analysis of patients' thoughts, which can potentially automate and assist all stages of the psychotherapy process. We start from CBT since it is the only therapy paradigm we found publically available dataset - However, we believe the cognition ability of LLMs we elicited, as well as the effectiveness of our method, can easily be extended to a wide range of therapy paradigms.
>
> Psychotherapy mostly involves the understanding and generation of speech and written content since it is conducted primarily based on communications between therapists and patients. It overlaps with language research in nature. Our paper demonstrates the great promise and vision for the new era of the research direction of NLP for psychotherapy. We believe our contribution meets the standard of a short paper in the submitted tracks “Theme Track: Large Language Models and the Future of NLP” and “NLP Applications.”
>
> Chain-of-Thought Prompting Elicits Reasoning in Large Language Models, Wei et al., 2022. In NeurIPS.
>
> ### 2. Regarding “Diagnosis" is kind of over-claim”
> In the proposed DoT framework, we demonstrate that GPT-3.5 and GPT-4 can generate diagnosis rationales with promising performance verified by human experts. Note that our method is intended to serve as assistance for psychotherapy professionals to perform diagnosis, but is never meant to replace the professionals. We enlighten the vision and promise that LLM can potentially generate diagnosis of patients’ thoughts. Regarding the mechanism as well as the robustness of such ability, how to further improve the performance, and how to build real-world systems to assist professionals, we believe they are all valuable future directions starting from this paper.

---

### Official Review · Reviewer_sC35 · 2023-08-05

**Soundness:** 3

**Excitement:**

4: Strong: This paper deepens the understanding of some phenomenon or lowers the barriers to an existing research direction.

**Paper Topic And Main Contributions:**

This paper proposes to use the Diagnosis of Thought (DoT) prompting with large language models (LLMs) to detect cognitive distortion types of patients for psychotherapy. The examples of cognitive distortion include suspecting that others are hiding intentions, taking up the blame for a situation, or looking at a situation in a black-and-white way. Detecting this cognitive distortion types could help both psychotherapists and patients. The proposed DoT method is to prompt an LLM with three questions for the three stages of (1) subjective assessment, (2) contrastive reasoning, and (3) schema analysis. In the subjective assessment stage, we identify objective facts and subjective thoughts in the patient's speech. The contrastive reasoning stage a provides reasoning  process to support or contradict the patient's subjective thoughts. In the schema analysis stage, we want to figure out the cognitive structure of the patient is relying on. The paper shows evaluations on a public cognitive distortion detection dataset, and show that the proposed prompting method is more effective than other methods (directly generating the results and zero-shot CoT prompting). The ablation study was performed to study the effect of including each stage. Then, human evaluation was also performed to assess the quality of generated rationales. These generated rationales provides interpretability/explainability to users when the system is deployed in field.

**Questions For The Authors:**

A. line 237: any details about the full-training model?

B. any details about the experts for human evaluation? How many experts assessed the outcome? Who are they? How did you compute the inter-agreement?

**Reasons To Accept:**

This paper presents an impressive illustration of how large language models can be applied effectively. The application addresses a compelling and significant issue related to psychotherapy, and the proposed prompting method is thoughtfully grounded in domain expertise.

**Reasons To Reject:**

While the lack of technical contributions may be a concern for this conference, personally, I don't view it as a problem for this paper.

**Reproducibility:**

2: Would be hard pressed to reproduce the results. The contribution depends on data that are simply not available outside the author's institution or consortium; not enough details are provided.

**Reviewer Confidence:**

4: Quite sure. I tried to check the important points carefully. It's unlikely, though conceivable, that I missed something that should affect my ratings.

---

> ### Author Rebuttal · Authors · 2023-08-28
>
> We thank the reviewer for the comments and support for this work.
>
> ### 1. Regarding “lack of technical contributions”
> Our proposed DoT framework comes from the consultation and study jointly with psychotherapy experts. The framework is concise and elegant - all stages in DoT address the key insights and challenges in diagnosing the patients’ thoughts without any redundancies. Each stage brings significant improvement for distortion assessment and classification, and the generated rationales are highly valuable for psychotherapy professionals to interpret, analyze, as well as learn more insights into the patients’ thought patterns. Given the critical condition of mental health issues and the severe shortage of therapists today, we believe it is imperative, or at least highly valuable, to call on efforts for NLP in psychotherapy, which can benefit billions of lives. This is one of the most important purposes of this work. We are happy that the reviewer acknowledges the vision and contribution of this work.
>
> ### 2. Answer to Question A:
> The full-training model is the best-performing one from Shreevastava et al., 2021 paper. They use sentence-BERT combined with the K-Nearest Neighbors method for classification.
>
> Sagarika Shreevastava and Peter Foltz. 2021. Detecting Cognitive Distortions from Patient-Therapist Interactions.
>
> ### 3. Answer to Question B:
> See our Ethics Statement section for details regarding the hired experts. Specifically, we hired psychotherapy experts from UpWork (www.upwork.com, a platform for hiring professionals). The experts we hired are certified clinicians, therapists, and counseling psychology Ph.D. students. They all have expertise in cognitive behavior therapy (the target therapy paradigm of cognitive distortion detection). We discussed with each of them to reach an agreement on the payment and signed the contract. The average hourly rate for all the experts is $80. We employ two experts for each evaluation - four experts in total (two for results of GPT-3.5 and two for GPT-4). The agreement rate is the percentage of examples the two experts gave the same rating. We will add more details to the Ethics Statement section in the revision.

---

### Official Review · Reviewer_eTZW · 2023-08-09

**Soundness:** 2

**Ethical Concerns:**

Yes

**Excitement:**

2: Mediocre: This paper makes marginal contributions (vs non-contemporaneous work), so I would rather not see it in the conference.

**Justification For Ethical Concerns:**

Ethical concerns:
The "experts" that were mentioned in the paper
Were there involved clinicians with expertise in "cognitive distortion detection"?
Did they were paid? How much did they were paid? or were they part of this research (co-authors)?
How they were found? who are they?

Because the paper deals with the field of mental health, it is worthwhile to involve experts from this field as part of the research or at least send the work to clinical reviewers too.

**Missing References:**

Cao, J., Tanana, M., Imel, Z. E., Poitras, E., Atkins, D. C., & Srikumar, V. (2019). Observing dialogue in therapy: Categorizing and forecasting behavioral codes. arXiv preprint arXiv:1907.00326.

Shwartz, V., West, P., Bras, R. L., Bhagavatula, C., & Choi, Y. (2020). Unsupervised commonsense question answering with self-talk.

Yao, S., Yu, D., Zhao, J., Shafran, I., Griffiths, T. L., Cao, Y., & Narasimhan, K. (2023). Tree of thoughts: Deliberate problem-solving with large language models. arXiv preprint arXiv:2305.10601.

Shapira, N., Levy, M., Alavi, S. H., Zhou, X., Choi, Y., Goldberg, Y., ... & Shwartz, V. (2023). Clever Hans or neural theory of mind? stress testing social reasoning in large language models.

**Paper Topic And Main Contributions:**

The authors used an existing labeled dataset from the field of Psychotherapy/CBT/cognitive-distortion-detection and classified the dataset with GPT4, GPT3.5, and Vicuna.
The authors suggested a new prompting mechanism with 3 stages, that extract information beyond the classification itself and use it to refine the classification. The authors also perform ablation tests and human evaluations for theIntermediate information extracted.

**Questions For The Authors:**

A: Are you releasing data or code?

B: How exactly did you measure the agreement between annotators? What were the exact guidelines for the experts? What was the procedure? to get an agreement?

C: How can you explain the results?  why GPT4+DoT is lower than GPT4 in Distortion assessment? and why GPT4+DoT is lower than GPT-4 in Distortion assessment but higher in Distortion Classification? What does it mean with respect to the interpretability?
Why ChatGPT+DoT is better in DA than DC but GPT4 results are opposite.

**Reasons To Accept:**

The results of the new models improve the performance of the task over a model trained in the previous literature.

The paper is written clearly and with good examples.

Although it is a small dataset (only 100 samples), human tagging work could possibly be useful in future works.

---

After reading the authors' response -
Thank you for answering.

I did miss the paragraph on ethics where it is described that the authors employed psychotherapists to label the data.

Indeed, I also missed that the authors sent the paper to "Theme Track: Large Language Models and the Future of NLP" or "Submission Track 2: NLP Applications", NLP Application sounds more appropriate.

I still think that more significant work needs to be done in related work (the authors agreed, but they did not provide such a paragraph, nor do I believe that the entire literature can be covered in such a short time)  and that the scientific innovation here is in the application and not in the method.
I still think that an expert from the field of cognitive-distortion-detection should be involved in building such an application (PhD students in psychology are not considered experts).
And if this is an application intended for clinicians, then clinicians should be part of the authors or\also clinicians should be involved in the review process.

**Reasons To Reject:**

There are flaws that are detailed below. The main concern is that the EMNLP stage is not the right place given the scientific innovation, lacking relevant related-work-literature-review, and the scope of the work.
Also, since the authors are building a system for psychotherapy usage, and the main innovation is for the psychotherapy domain, this type of work must be assessed by clinicians too, I highly recommend to re-submit this work to the CLPsych workshop within ACL (that contains reviewers from both domains psychotherapy and NLP and also can tolerate low scientific innovation).

line 46 "However, existing works mostly take shallow attempts in a heuristic manner, e.g., analyzing emotions and generating comforting responses."

There are several researchers who work with technological startups that combine psychotherapy experts with advanced automated methods among them:
Zac Imel (Lyssn.io), Jonathan D. Huppert (expert in CBT and researching the field of automated tasks), Shiri Sadeh-Sharvit (Eleos Health)

see also
Cao, J., Tanana, M., Imel, Z. E., Poitras, E., Atkins, D. C., & Srikumar, V. (2019). Observing dialogue in therapy: Categorizing and forecasting behavioral codes. arXiv preprint arXiv:1907.00326.


line 68: "For the psychology domain, ChatGPT and GPT-4 present very promising performance in the classic Sally-Anne test (Baron-Cohen et al., 1985; Bubeck et al., 2023)"

Bubeck et al (2023) paper received severe criticism from the scientific community and in particular on the subject of Sally-Anne test (Baron-Cohen et al., 1985)
See:
Shapira, N., Levy, M., Alavi, S. H., Zhou, X., Choi, Y., Goldberg, Y., ... & Shwartz, V. (2023). Clever Hans or neural theory of mind? stress testing social reasoning in large language models.

See also the section "Artificial intelligence" under "Sally-Anne test" in Wikipedia, they relate to Bubeck et al (2023) claim and cite more scientific references:
https://en.wikipedia.org/wiki/Sally%E2%80%93Anne_test


The scientific innovation here does not come from the field of interpretability nor from the field of NLP (generally known methods are used to perform classification, there is nothing new here), the innovation is more suitable for the psychotherapy community. I recommend submitting the work to the CLPsych workshop.

The connection to the field of interpretability is weak. There is no reference to the reliability of the interpretation. there are no references at all for the interpretability-related work.  How is it different/similar to previous works?

The literature review and related work is weak (both from the psychotherapy research and interpretability or NLP).

line 230:
"For Vicuna, both ZCoT and DoT exceed the token limits for many examples; we omit the results."
how many samples have you omitted? did you score them as wrong answers for the evaluation?

It is worth mentioning already in the introduction that this is not a new task but an improvement of results using GPT4 on an existing task that has already been published in Kagel earlier.

Where are the ablation studies for GPT4

In human evolution, you proved that the explanations make good sense.
But you did not prove that there is a connection between the explanations and the results of the final classification.

**Reproducibility:**

3: Could reproduce the results with some difficulty. The settings of parameters are underspecified or subjectively determined; the training/evaluation data are not widely available.

**Reviewer Confidence:**

4: Quite sure. I tried to check the important points carefully. It's unlikely, though conceivable, that I missed something that should affect my ratings.

**Typos Grammar Style And Presentation Improvements:**

Typos and suggestions
* The font in Table 1 is too small, making it hard to read
* Table 5 "o, or" --> ", or"
* Appendix B could be written in a clearer way (using a table, explaining what is the blue lines and what are the rest)
* Appendix C's content is missing (I guess you meant that it includes Figure 3 (?) write it also in the text.
* Add the annotator's label for the examples you added in Appendix C.

---

> ### Author Rebuttal · Authors · 2023-08-28
>
> We thank the reviewer for the comments and suggestions. We would like to respond to your concerns as follows:
>
> ### 1. Regarding “EMNLP stage is not the right place… ”
> We believe the interdisciplinary nature, as well as the nature of creating/exploring new research directions, should not be the reason to block research work from major conferences and only limit it to workshops.
>
> As we demonstrate in the introduction, mental health condition worldwide has been growing extremely critical in recent years. Given the severe shortage of therapists and the verbal/textual nature of psychotherapy, we believe NLP for psychotherapy should be one of the most important next research focuses to benefit billions of lives. NLP for psychotherapy hasn’t been a popular research field in the past due to its challenge, and hasn’t established a large community like other interdisciplinary AI fields (e.g., AI for drug synthesis). And now, with the promising ability shown by LLMs, we believe it’s the correct time to raise attention and efforts in this direction to significantly strengthen this new community. This is one of the most important purposes of this work. The impact of workshops is far smaller than major conferences. We believe we should not limit valuable new fields to workshops that slow down their growth.
>
> It is not accurate to say, “The main innovation is for the psychotherapy domain.” Our proposed DoT framework targets deep analysis of the patient's speech, involving generating entailments/contradictions, commonsense understanding, emotion understanding, and schema summarization, covering a wide range of tasks in the language analysis/generation domain.
> Nonetheless, for any interdisciplinary research direction, it is hard and also not beneficial to draw a rigid boundary between the knowledge from two sides. As mentioned above, psychotherapy mostly involves the understanding and generation of speech and written content. It overlaps with language research in nature. We believe new directions and diversities should be embraced in order for NLP to thrive in the new era.
>
> ### 2. Regarding “this type of work must be assessed by clinicians too,”
> in section 3, we stated that the DoT framework is established upon consultation and discussion jointly with psychotherapy professionals. Also, please see our Ethics Statement section for details regarding hiring psychotherapy experts. Specifically, we hired psychotherapy experts from UpWork (www.upwork.com, a platform for hiring professionals). The experts we hired are certified clinicians, therapists, and counseling psychology Ph.D. students. They all have expertise in cognitive behavior therapy (the target therapy paradigm of cognitive distortion detection). We discussed with each of them to reach an agreement on the payment and signed the contract. The average hourly rate for all the experts is $80. The involvement of all these experts in this work guarantees the high quality, correctness, as well as reliability of the domain expertise.
>
> ### 3. Regarding Bubeck et al. (2023) paper on the Sally-Annie test
> Shapira et al. (2023) paper concludes that “LLMs demonstrate enhanced, but not robust theory-of-mind ability.” In our work, we are inspired by such enhanced ability and investigate the application of diagnosing patients’ thoughts in psychotherapy. Our experiments on the real dataset (not anecdotal examples) show very promising performance both in automatic and human expert evaluation. Whether LLM can learn the general theory-of-mind ability and the mechanism of learning are challenging research questions in themselves - we would like to leave them for future work. Shapira et al. (2023) paper was published only one month before the EMNLP deadline, and we had not read it by that time. We will add more discussions regarding this in the revision.
>
> ### 4. Regarding “lacking relevant related-work-literature-review… ”
> Due to the space constraint of the short paper, we mostly discuss the related work in the introduction of the current version. We will add a related work section and discuss all the papers you mentioned on the extra page if accepted.
>
> ### 5. Regarding “The connection to the field of interpretability”
> We would like to clarify that this paper is submitted to the tracks “Theme Track: Large Language Models and the Future of NLP” and “NLP Applications,” not the “Interpretability for NLP” track. In our paper, we use the term “Interpretability” to describe the benefits of the generated diagnosis rationales - with such rationale, the diagnosis process and the results can be interpreted and verified by human experts, as well as providing assistance to the patients to do thought analysis, as stated in section 3.
>
> ### 6. Regarding Vicuna results
> For ZCoT and DoT, over 95% of examples go beyond the Vicuna token limit (2048) before generating the DA and DC results. So we remove these two experiments for Vicuna and only present the results for vanilla Vicuna zero-shot performance (directly generating the DA and DC results).
>
> ### 7. Regarding “an improvement of results using GPT4 on an existing task… ”
> For the methodology part of this work, our contribution is the DoT prompting, which can be applied to any advanced LLM with sufficient input length. We use GPT-3.5 and GPT-4 as underlying LLM in our experiments.
>
> ### 8. Regarding “ablation studies for GPT4”, “connection between the explanations and the results of the final classification”
> We demonstrate major improvements in both distortion assessment (DA) and distortion classification (DC) for all three stages of DoT for GPT-3.5. For GPT-4, the improvement is marginal. One possibility is data contamination - the dataset we use is publicly released, so GPT-4 may have already seen it in its training data and achieved a near upper-bound performance no matter with DoT or not. With generating explanations for all three DoT stages, we demonstrate significant improvements for GPT-3.5. Also, the good quality of the explanation itself is very valuable for assisting psychotherapy, as we stated in section 3.
>
> ### 9. Answer to Question A
> Yes, we will release all our data and code upon acceptance.
>
> ### 10. Answer to Question B
> We employ two experts for each evaluation (the cost of recruiting psychotherapy experts is high). The agreement rate is the percentage of examples the two experts gave the same rating. Regarding the guidelines for the experts, section 4.3 describes our evaluation procedure. For each hired expert, we first present the task instructions, have discussions, and ask them to practice a few examples to make sure they understand the task. Specifically, for each example, we present the patient’s speech, all the prompts, and the generated rationales for all stages. We ask the expert to rate the rationale generated for each stage based on the rating criteria described in section 4.3. We will add more details regarding the guidelines in the revisions.
>
> ### 11. Answer to Question C
> Assessing the distortion is easier than the more fine-grained classification, also with less variance. So, the improvement in DA is greater than DC for ChatGPT+DoT. However, for GPT-4, as we explained previously, we did not observe a significant improvement for DA and DC using GPT-4+DoT, due to the possibility of data contamination. In this case, the addition of DoT will possibly introduce more interference into the “pattern” that GPT-4 memorized in data contamination, leading to slight performance decrease and instability. But for DC, since it’s more challenging and variant than DA, DoT still brings slight improvement.
>
> ### 12. Regarding Typos and suggestions
> Yes, Figure 3 is the content of Appendix C. We will address all your suggestions in the revisions.
>
> ### 13. Regarding Ethical concerns
> Please see our Ethics statement section, as well as our response point #2 for details of hiring psychotherapy experts.

---

### Meta-Review · Area_Chair_Xvbr · 2023-09-16

**Recommendation:** 3

**Metareview:**

This paper examines the automatic identification of cognitive distortions with “diagnosis of thought” prompting. They use a three stage technique: first assessing the subjectivity of statements, second looking at statements that support or contradict the statements, and lastly an analysis of schemas, which means matching the statements to a cognitive distortion type. They show that using ChatGPT and GPT-4 they can improve distortion detection with this prompting approach. The two of the reviewers are not excited by this work, seemingly because of the novelty of the prompting method. The application area is exciting and does have potentially high impact. Reviewer eTZW makes some helpful suggestions for related work that should be expanded upon. The human evaluation and consultation with experts is valuable in this paper. I disagree with reviewers in that their perceived lack of novelty should not lead to a low soundness score, so I think this paper is appropriate for findings.

---

### Decision · Program_Chairs · 2023-10-07

**Decision:**

Accept-Findings

**Comment:**

This paper examines the automatic identification of cognitive distortions with “diagnosis of thought” prompting. They use a three stage technique: first assessing the subjectivity of statements, second looking at statements that support or contradict the statements, and lastly an analysis of schemas, which means matching the statements to a cognitive distortion type. They show that using ChatGPT and GPT-4 they can improve distortion detection with this prompting approach. The two of the reviewers are not excited by this work, seemingly because of the novelty of the prompting method. The application area is exciting and does have potentially high impact. Reviewer eTZW makes some helpful suggestions for related work that should be expanded upon. The human evaluation and consultation with experts is valuable in this paper. I disagree with reviewers in that their perceived lack of novelty should not lead to a low soundness score, so I think this paper is appropriate for findings.